mathematical modelling/energy/chemical engineering

zinc–air battery, dynamic model, linear parameter-varying model, nonlinear model, linear model

**Author for correspondence:**
Soorathep Kheawhom
e-mail: soorathep.k@chula.ac.th

This article has been edited by the Royal Society of Chemistry, including the commissioning, peer review process and editorial aspects up to the point of acceptance.

# Linear parameter-varying model for a refuellable zinc–air battery

Woranunt Lao-atiman[1], Sorin Olaru[4], Sette Diop[4], Sigurd Skogestad[5], Amornchai Arpornwichanop[1,2], Rongrong Cheacharoen[3] and Soorathep Kheawhom[1,2]

[1]Department of Chemical Engineering, Faculty of Engineering, [2]Center of Excellence in Process and Energy Systems Engineering, and [3]Metallurgy and Materials Science Research Institute, Chulalongkorn University, Bangkok, Thailand
[4]CentraleSupélec, Laboratoire des signaux et systèmes, Université Paris-Saclay, CNRS, Gif sur Yvette, France
[5]Department of Chemical Engineering, Norwegian University of Science and Technology, Trondheim, Norway

SK, 0000-0002-3129-2750

Due to the increasing trend of using renewable energy, the development of an energy storage system (ESS) attracts great research interest. A zinc–air battery (ZAB) is a promising ESS due to its high capacity, low cost and high potential to support circular economy principles. However, despite ZABs' technological advancements, a generic dynamic model for a ZAB, which is a key component for effective battery management and monitoring, is still lacking. ZABs show nonlinear behaviour where the steady-state gain is strongly dependent on operating conditions. The present study aims to develop a dynamic model, being capable of predicting the nonlinear dynamic behaviour of a refuellable ZAB, using a linear parameter-varying (LPV) technique. The LPV model is constructed from a family of linear time-invariant models, where the discharge current level is used as a scheduling parameter. The developed LPV model is benchmarked against linear and nonlinear model counterparts. Herein, the LPV model performs remarkably well in capturing the nonlinear behaviour of a ZAB. It significantly outperforms the linear model. Overall, the LPV approach provides a systematic way to construct a robust dynamic model which well represents the nonlinear behaviour of a ZAB.

# 1. Introduction

Renewable energy has great potential to sustain global energy security. Nevertheless, renewable energy is very intermittent

and highly erratic, resulting in fluctuation in energy production. An energy storage system (ESS) can stabilize such fluctuation and effectively support energy management and integration. Recently, ESS has become an immensely focused topic in energy research. An ESS can enhance the efficiency and stability of various energy systems [1,2] (table 1).

Of the various types of ESS, zinc–air batteries (ZABs) prove to be the most promising, providing excellent specific capacity. ZAB technology has made substantial research progress and is approaching commercialization [3–5]. ZABs use the electrochemical reaction between zinc (Zn) and oxygen ($O_2$) to store and release electricity. ZABs characteristically have high energy density but low power. It is reported that ZABs are able to deliver peak power density up to 430 mW cm$^{-2}$ and energy density up to 837 W h kg$^{-1}$ [6]. These values have already exceeded the specific energy of commercialized lithium ion batteries (LIBs) many times. Moreover, Zn is abundant on Earth; therefore, its cost is quite low [7–9]. In addition, Zn is safe, environmentally friendly and highly stable. Zinc oxide (ZnO), which is the discharge product, can be easily recycled. $O_2$, supplied from atmospheric air, is also quasi-free and virtually unlimited. Thus, ZABs present great potential and feasibility in providing a decent ESS on a large scale.

Generally, a ZAB consists of two electrodes: a Zn electrode (negative electrode) and an air electrode (positive electrode). The most common electrolyte for a ZAB is an aqueous alkaline electrolyte such as potassium hydroxide (KOH) solution. As regards discharging, Zn serves as an electron donor at the negative electrode. Zn reacts with hydroxide ions (OH$^-$) producing zincate ions (Zn(OH)$_4^{2-}$) and electrons (e$^-$). Zincate ions remain in the electrolyte and can precipitate to form ZnO. At the positive electrode, $O_2$ from the ambient air acts as an electron acceptor. Oxygen reduction reaction (ORR) proceeds and provides OH$^-$ as the product. The overall reaction of a ZAB is the redox reaction of Zn and $O_2$, thereby producing ZnO. The overall reactions that occur in the battery are described as follows [10,11]:

$$\text{Negative electrode:} \quad \text{Zn} + 4\text{OH}^- \leftrightarrow \text{Zn(OH)}_4^{2-} + 2\text{e}^-$$

$$\text{Zn(OH)}_4^{2-} \leftrightarrow \text{ZnO} + 2\text{OH}^- + \text{H}_2\text{O}$$

$$\text{Positive electrode:} \quad \frac{1}{2}\text{O}_2 + \text{H}_2\text{O} + 2\text{e}^- \leftrightarrow 2\text{OH}^-$$

and

$$\text{Overall reaction:} \quad \text{Zn} + \frac{1}{2}\text{O}_2 \leftrightarrow \text{ZnO}.$$

The theoretical open circuit voltage (OCV) is approximately 1.65 V [10], which can be calculated from the following equation:

$$E_{0,\text{cell}} = \left( E_{0,\text{air}} + \frac{RT}{n_e F} \ln \frac{[\text{O}_2]^{0.5}}{[\text{OH}^-]^2} \right) - \left( E_{0,\text{Zn}} + \frac{RT}{n_e F} \ln \frac{[\text{Zn(OH)}_4^{2-}]}{[\text{OH}^-]^4} \right), \tag{1.1}$$

where $E_{0,\text{cell}}$ is the standard cell potential or theoretical OCV, $E_{0,\text{air}}$ is the standard electrode potential of the air electrode (corresponding to ORR) which is 0.401 V versus standard hydrogen electrode (SHE), $E_{0,\text{Zn}}$ is the standard electrode potential of Zn electrode (corresponding to Zn oxidation reaction) which is −1.26 V versus SHE, $R$ is the gas constant, $T$ is the temperature, $n_e$ is the number of electron transfers in the reaction and $F$ is the Faraday constant. This equation uses the concentration of the reactants to calculate the standard cell potential.

However, the practical OCV obtained from laboratory prototypes is about 1.4 V [12–14]. Charging can be done in a rechargeable ZAB by applying a potential higher than the theoretical OCV. When charging, the reactions proceed backwards and regenerate Zn and $O_2$.

The development of a ZAB encompasses many aspects [15–18]. In the past decade, the focus has been on improving the performance and stability of the battery such as development of ORR catalyst or battery electrolyte. It is noted that the performance of a ZAB has been improved by optimizing battery parameters [19]. The development of battery operation, i.e. pulse-current charging, has also been investigated. Pulse-current charging is a technique developed to prevent the growth of dendritic zinc when charging the battery [20–22]. While most research concentrates on the improvement of material and battery design, management and monitoring tools for a ZAB have received less attention and clearly represent an incomplete field of study. Management systems can improve the performance of batteries and protect batteries from inappropriate operations [23,24]. For instance, when ZABs are charged with excessive voltage, both the detrimental dendritic formation and hydrogen evolution reaction (HER) occur. Management systems require precise prediction of dynamic behaviour and state

**Table 1.** Nomenclature.

| | |
|---|---|
| $A$ | state matrix in state space model |
| $B$ | input matrix in state space model |
| $BC$ | combined parameter between parameters $B$ and $C$ |
| $B/F$ | linear block in Hammerstein–Wiener (HW) model |
| $B(z)$ | numerator polynomial function of linear block in HW model |
| $b_{nb}$ | polynomial coefficient of $B(z)$ |
| $C$ | output matrix in state space model |
| $C_P$ | capacitance in RC loop, F |
| $D$ | feedthrough matrix in state space model |
| $E_{0,air}$ | standard electrode potential of air electrode, 0.401 V versus standard hydrogen electrode (SHE) |
| $E_{0,cell}$ | standard cell potential or theoretical OCV, V |
| $E_{0,zn}$ | standard electrode potential of Zn electrode, −1.26 V versus SHE |
| $F$ | Faraday constant, 96485.3329 A s mol$^{-1}$ |
| $F(z)$ | denominator polynomial function of linear block in HW model |
| $f$ | input nonlinear block |
| $f_{nf}$ | polynomial coefficient of $F(z)$ |
| $h$ | output nonlinear block |
| $I_{cell}$ | discharge current, A |
| $k$ | discrete time, s |
| $n$ | number of states $= 1$ |
| $n_b$ | order of $B(z)$ polynomial |
| $n_e$ | number of electron transfer in the reaction |
| $n_f$ | order of $F(z)$ polynomial |
| $n_k$ | input delay of linear block in HW model |
| $p$ | scheduling parameter |
| $R$ | gas constant, 8.3145 J mol$^{-1}$ K$^{-1}$ |
| $R_C$ | resistance in RC loop, $\Omega$ |
| $R_0$ | ohmic resistance, $\Omega$ |
| $T$ | temperature, K |
| $T_s$ | sampling time, s |
| $u$ | input vector of state-space model and HW model |
| $V_{OC}$ | open circuit potential, V |
| $V_{RC}$ | potential loss, V |
| $V_{RCR}$ | potential drop across RC loop, V |
| $w$ | input of linear block in HW model |
| $X$ | state vector in state-space model |
| $x$ | output of linear block in HW model |
| $Y$ | output vector of state-space model |
| $y$ | output of HW model |
| $z$ | delay operator in output-error model |
| $\alpha$ | coefficient of two-term exponential function |
| $\beta$ | coefficient of two-term exponential function |
| $\gamma$ | coefficient of two-term exponential function |
| $\delta$ | coefficient of two-term exponential function |
| $\mu$ | coefficient of third-order polynomial function |
| $\xi$ | model parameter estimated from correlations |

of the battery, which is typically achieved via modelling. Some types of modelling have been used in ZAB researches. As such, theoretical continuum models have been carried out and used to examine phenomena occurring inside the battery [20,25,26].

The dynamic behaviour of a battery focuses on the discharge current and voltage of the battery, which is considered as being the input and output of the system. Thus, empirical modelling has regularly been preferred, due to its simplicity in computation. For example, an equivalent circuit model (ECM) is the most commonly used empirical model in the investigation of battery dynamics. An ECM describes the dynamic behaviour of the battery via simple electrical elements that are comparable to the electrochemical characteristics of the battery [27]. This type of model has been used in various batteries, such as LIBs [28–30], Zn–Ni batteries [31,32] or lead–acid batteries [33]. However, only a few works on a ZAB have used ECM to predict battery behaviour [34], although electrochemical impedance spectroscopy (EIS) has frequently been applied. For a more empirical approach, a state-space model has been developed. This model is normally used with both state and parameter estimation algorithms [27].

Although the dynamic behaviour of a ZAB is strongly nonlinear, previous studies have centred on the development of empirical linear models. Nonlinear behaviour can be realized by invoking first principles-based models or nonlinear empirical modelling techniques. However, it is acknowledged that nonlinear models are less flexible than comparable linear models and the mathematical tools are lacking for nonlinear systems. Alternatively, nonlinear behaviour can be captured via a linear parameter-varying (LPV) model, which approximates a nonlinear system with high accuracy [35,36].

LPV models have been applied in various systems, but only a few works have employed this technique in a battery system [37–39]. For instance, a subspace method has been introduced for the identification of an LPV battery model for LIBs, where state of charge (SOC) estimation was done using LPV techniques [37,38]. Results indicated that this technique provides good and stable performance and is easy to tune compared with other algorithms. In another example, LPV modelling has been used to assist in monitoring the state of health (SOH) for an LIB cell [39]. This model combined with a nonlinear Kalman filter proved capable of online estimating SOC and SOH. The model was validated via measurement data and provided good validation results.

Herein, an LPV model is developed to account for all nonlinearities within a ZAB directly. Nonlinear ZAB characteristics, therefore, are empirically exhibited in the form of change in parameters of the underlying linear time-invariant (LTI) models, with respect to a reference condition. The LPV model is seen to combine the varying parameters into a single model. Besides, it proved capable of effectively predicting battery nonlinear behaviour over a wide range of conditions. Furthermore, the LPV model adopted the linear characteristic of the LTI model. Hence, it possessed considerable robustness.

This work proposes to use LPV models for predicting the input–output discharge behaviour of a ZAB. Data employed in this scheme were obtained from an in-house refuellable ZAB [40]. The underlying linear models obtained at different conditions are then combined into a single LPV model, where the discharge current level is used as a scheduling parameter. As regards validation, the developed LPV model is used to predict various sets of response data. A nonlinear model was further implemented to compare results between the nonlinear and LPV model.

# 2. Battery description and experimental data

Battery response data previously published by Lao-atiman *et al.* [40] have been implemented for parameter estimation and model validation. As shown in figure 1, such data were acquired from a tubular refuellable ZAB, designed in-house. The cylindrical structure of the cell was made of stainless-steel mesh. The active material for the anode was 6 g of 20 mesh Zn pellets packed into another stainless-steel mesh tube. The cathode current collector comprised nickel (Ni) foam coated with ORR catalyst ($MnO_2$) and a gas diffusion layer. The cell contained 8 M KOH aqueous solution as the electrolyte.

After battery fabrication, both the discharge current and voltage of the battery were measured by BA500 battery analyser (Battery Metric, Toronto, ON, Canada). Sampling time was 1 s. Then, the discharge current setpoint was set. Subsequently, the battery was forced to discharge in accordance with the setpoint. Next, both the actual discharge current and voltage were measured and recorded along with the selected sampling time. The set of data used for model identification contained a time-series of discharge current (as input) and discharge voltage (as output). The discharge voltage was measured at the specified discharge current. Step response data, including the discharging current steps from 0 to 100, 0 to 450 and 0 to 900 mA, were used to identify linear models. With respect to

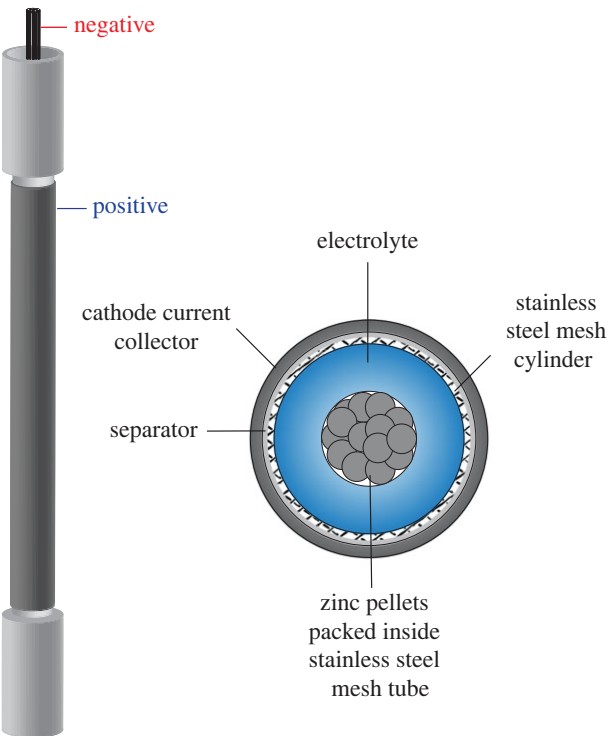

**Figure 1.** Schematic diagram of experimental ZAB.

validation, response data with increased variability and complexity were examined. All data used in this work, including data names and descriptions, are summarized in table 2. Graphical representation for each dataset can be found in electronic supplementary material, figures S1–S10.

# 3. Methodology

The LPV model is a collection of LTI state-space models whose parameters vary as a function of scheduling parameters. In the case of a ZAB, discharge current is considered to be the scheduling variable, which is available for measurement. More importantly, the discharge current is the signal which directly enables modifications of a ZAB's dynamic behaviour to occur.

In terms of methodology, this approach follows a classical operation mode: namely, a certain number of points in the scheduling space were selected. Thus, an LTI model was constructed and assigned to each point, representing the dynamics in the local vicinity of that point. The dynamics at scheduling locations in between the specified grid points were acquired by interpolation of LTI models at neighbouring points.

In addition, a nonlinear model was constructed to benchmark the LPV model in terms of precision and complexity of the prediction.

## 3.1. Linear state-space model

The LPV model uses local LTI models: the choice being made here is to represent these models in a discrete domain, taking into account that available data are inherently obtained on a discrete timescale after sampling. Trajectories of a state vector $(X)$ and output vector $(Y)$ are commonly measured and tracked as they move through time. The LTI model, at each local operation point, is expressed as in the below equations

$$X(k + 1) = AX(k) + Bu(k) \tag{3.1}$$

and

$$Y(k) = CX(k) + Du(k), \tag{3.2}$$

where $u$ is an input vector. As for a single-input, single-output case,

**Table 2.** Summary of experimental data used for identification and validation of models.

| data name | description | description |
|---|---|---|
| 0T100 | current step from 0 to 100 mA | StepDischarge.xlsx [40] sheet: 100STEP0-100-0 |
| 100T0 | current step from 100 to 0 mA | StepDischarge.xlsx [40] sheet: 100STEP0-100-0 |
| 0T450 | current step from 0 to 450 mA | StepDischarge.xlsx [40] sheet: 450STEP0-450-0 |
| 450T0 | current step from 450 to 0 mA | StepDischarge.xlsx [40] sheet: 450STEP0-450-0 |
| 0T900 | current step from 0 to 900 mA | StepDischarge.xlsx [40] sheet: 900STEP0-900-0 |
| 900T0 | current step from 900 to 0 mA | StepDischarge.xlsx [40] sheet: 900STEP0-900-0 |
| 400T500R | repeating current step between 400 and 500 mA | StepDischarge.xlsx [40] sheet: 100STEP400-500 |
| 500T1000R | repeating current step between 500 and 1000 A | StepDischarge.xlsx [40] sheet: 500STEP500-1000 |
| MULTI | multiple current step from 0 to 100, 450 and 900 mA | Supplementary.xlsx[a] sheet: MULTI |
| VARIOUS | various current step with random pattern | Supplementary.xlsx[a] sheet: VARIOUS |

[a]The data are located in the electronic supplementary material.

$$A \in \mathbb{R}^{n \times n},\ B \in \mathbb{R}^{n \times 1},\ C \in \mathbb{R}^{1 \times n}\ \text{and}\ D \in \mathbb{R}$$

$A$, $B$, $C$ and $D$ matrices are estimated from the experimental data ($Y(k)$, $u(k)$) via least square regression.

In this case, the input and output of the experimental data are discharge current and cell voltage, respectively. For convenience of computation, $Y$ represents the deviation of cell voltage from the OCV (potential loss). Then, $u$ represents the discharge current. This change of coordinate ensures that both $Y$ and $u$ are expressed in the absence of excitation and have a fixed point at 0, according to the LTI model, as shown in equations (3.1) and (3.2).

## 3.2. LPV model

As regards the LPV model, system dynamics are represented as a linear state-space model having parameters expressed in terms of functions of $r$ scheduling variables [41]. The case of a single-input, single-output system is denoted as follows:

$$A{:}\mathbb{R}^{r} \to \mathbb{R}^{n \times n},\ B{:}\mathbb{R}^{r} \to \mathbb{R}^{n \times 1},\ C{:}\mathbb{R}^{r} \to \mathbb{R}^{1 \times n}\ \text{and}\ D{:}\mathbb{R}^{r} \to \mathbb{R}.$$

The LPV model is a generalization of the LTI structure, building on the principles that dynamic properties vary with respect to the functioning conditions (represented by exogenous or internal signals) or parameters. Explicitly, model parameters are a function of the scheduling vector of parameters $p$ which in turn is time-varying

$$A = A(p(k)),\ B = B(p(k)),\ C = C(p(k))\ \text{and}\ D = D(p(k)). \tag{3.3}$$

Accordingly, the state-space model becomes

$$X(k + 1) = A(p(k))X(k) + B(p(k))u(k) \tag{3.4}$$

and

$$Y(k) = C(p(k))X(k) + D(p(k))u(k). \tag{3.5}$$

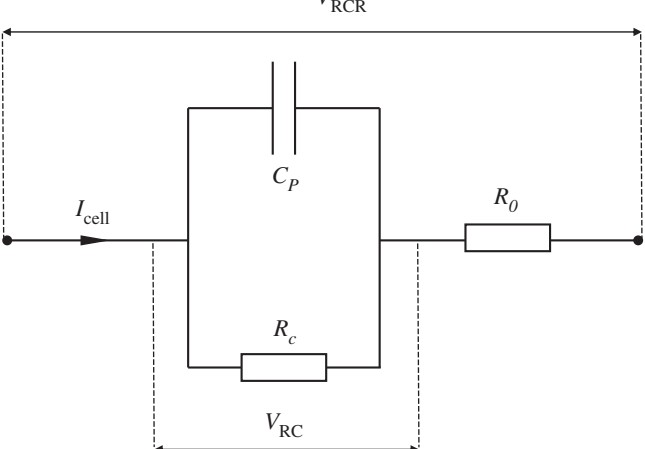

**Figure 2.** Electrical equivalent circuit diagram of potential loss of battery: first-order RC model.

For simplicity of notation, in time, the time dependence of the parameter will be dropped. With respect to ZAB modelling, given the fact that the experiments are conducted at constant external temperature, it is, therefore, assumed that the parameters are concentrated in the discharge current.

An important remark related to the particular single-input single-output form is that parametric dependence in both $B$ and $C$ has a certain degree of redundancy, as long as it relates to the input–output gain in equations (3.1) and (3.2) and can lead to non-unicity problems. To solve this issue and simplify equations (3.4) and (3.5), the coefficients of the matrix $C$ are assumed to be time-independent and considered constant through an appropriate change of coordinate leading to the form, as shown in equations (3.6) and (3.7)

$$X(k+1) = A(p)X(k) + B(p)u(k) \tag{3.6}$$

and

$$Y(k) = CX(k) + D(p)u(k). \tag{3.7}$$

The state-space model becomes more useful as the number of parameters are reduced. This form of model can also be interpreted as first-order resistor–capacitor (RC) model, as illustrated in figure 2.

In the circuit, there is an RC loop ($R_C$ and $C_P$) connected with another resistor ($R_0$). In a comparison between the state-space model and the equivalent circuit, it was found that the input, $u$, is equivalent to the discharge current ($I_{cell}$). Output, $Y$, is equivalent to $V_{RCR}$ which is potential loss of battery. The state variable, $X$, can be interpreted as the potential drop across the RC loop ($V_{RC}$). Parameter $A$ is equivalent to $1 - T_s/R_C C_P$. Parameters $B$ and $D$ are $T_s/C_P$ and $R_0$, respectively. Parameter $C$ equals to 1 which agrees with the assumption previously made. The state-space model can be rewritten as ECM, as follows:

$$V_{RC}(k+1) = \left(1 - \frac{T_s}{R_C C_P}\right)V_{RC}(k) + \frac{T_s}{C_P}I_{cell}(k) \tag{3.8}$$

and

$$V_{RCR}(k) = V_{RC}(k) + R_0 I_{cell}(k). \tag{3.9}$$

For physical interpretation, ECM is normally used for investigating battery behaviour via EIS. Herein, the RC loop contributed to potential loss due to the electrochemical reactions: so-called activation overpotential. This overpotential is the potential required to drive the reactions viz. Zn oxidation and ORR for discharging the ZAB. Several researches have suggested that the overpotential strongly depends on the discharge current level and can be theoretically described by the Butler–Volmer approach [15,25,42]. Next, $R_0$ contributed to the potential loss to internal resistance: so-called ohmic overpotential. This loss increases proportionally with the current drawn from the battery.

As regards battery modelling, scheduling parameters can be chosen from various parameters. In this work, input–output behaviour depends on the level of discharge current. Therefore, the sets of parameters used for constructing the LPV model were obtained from the data having different discharge current conditions.

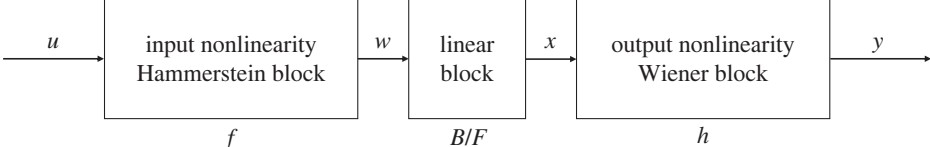

**Figure 3.** Block diagram of the Hammerstein–Wiener model.

As regards model parameters, the correlation between the model parameters (coefficients of the matrices in equations (3.6) and (3.7)) and discharge current was estimated via forms such as quadratic polynomial or exponential function

$$\text{quadratic polynomial:} \quad \xi = \mu_1 p^2 + \mu_2 p + \mu_3 \tag{3.10}$$

and

$$\text{two-term exponential:} \quad \xi = \alpha e^{\beta p} + \gamma e^{\delta p}, \tag{3.11}$$

where $\xi$ denotes the estimated parameter. $\mu_1, \mu_2$ and $\mu_3$ are the parameters acquired from the curve fitting. $\alpha, \beta, \gamma$ and $\delta$ are the coefficients of the exponential function obtained from the curve fitting.

The parameters of the linear model from the previous section were used to build the correlations with respect to after-stepping current levels, as scheduling parameters. From the experimental data, the conditions of current levels used for constructing the correlations were 0, 100, 450 and 900 mA. The correlations of parameters $A$, $B$ and $C$ were estimated by a second-order polynomial function. For parameter $C$, a linear function was used. An exponential function was used to fit the correlation of parameter $BC$. Correlations for parameters of the LPV model are provided in electronic supplementary material, table S4.

## 3.3. Nonlinear model

The nonlinear model used for comparison in this work was the Hammerstein–Wiener (HW) model. The HW model is a block-oriented model which contains nonlinear functions and a linear block separately [43,44]. As shown in figure 3, the HW model is depicted as a series of three connected blocks.

The first and last blocks are nonlinear functions which transform the input and output signals, respectively. The second block located between the two nonlinear blocks is the linear block. The first nonlinear block is called 'Hammerstein block' and is represented by function $f$, as shown in equation (3.12). This nonlinear block transforms the input signal before entering the linear block

$$w(k) = f[u(k)], \tag{3.12}$$

where $u(k)$ and $w(k)$ are the input and output of nonlinear block $f$, respectively.

The next block is the linear block and is denoted by $B/F$. The linear block is derived from an output-error (OE) model and transfers input $w(k)$ to output $x(k)$, as in the below equation

$$x(k) = \left(\frac{B}{F}\right) w(k - n_k), \tag{3.13}$$

where $n_k$ is an input delay. $B$ and $F$ are polynomials in a linear output-error model with respect to the delay operator $z^{-1}$ and defined, as follows in the below equations

$$B(z) = b_1 + b_2 z^{-1} + \ldots + b_{n_b} z^{-n_b+1} \text{ for } B \text{ order } = n_b \tag{3.14}$$

and

$$F(z) = 1 + f_1 z^{-1} + \ldots + f_{n_f} z^{-n_f} \text{ for } F \text{ order } = n_f. \tag{3.15}$$

The last nonlinear block $h$ is called the 'Wiener block'. This block transforms the output signal of the linear block, as in the below equation

$$y(k) = h[x(k)], \tag{3.16}$$

where $y(k)$ is the output of the nonlinear block $h$ and the output of HW model.

The output of the HW model $y(k)$ can be rewritten as a function of $u(k)$, as in the below equation

$$y(k) = h\left[\left(\frac{B}{F}\right) f[u(k)]\right]. \tag{3.17}$$

For this study, only the Hammerstein nonlinear block was used. The HW model, therefore, is reduced to the Hammerstein model.

**Table 3.** Summary of conditions for identification of the model used in this work.

| model name | model type | identification data[a] | identifying condition |
|---|---|---|---|
| SS0T100A | linear model | 0T100A | first-order model with feedthrough and |
| SS0T100B | linear model | 0T100B | 1 s sampling time |
| SS0T100C | linear model | 0T100C | number of states ($n$) $= 1$ |
| SS100T0A | linear model | 100T0A | |
| SS100T0B | linear model | 100T0B | |
| SS100T0C | linear model | 100T0C | |
| SS0T450A | linear model | 0T450A | |
| SS0T450B | linear model | 0T450B | |
| SS0T450C | linear model | 0T450C | |
| SS450T0A | linear model | 450T0A | |
| SS450T0B | linear model | 450T0B | |
| SS450T0C | linear model | 450T0C | |
| SS0T900A | linear model | 0T900A | |
| SS0T900B | linear model | 0T900B | |
| SS0T900C | linear model | 0T900C | |
| SS900T0A | linear model | 900T0A | |
| SS900T0B | linear model | 900T0B | |
| SS900T0C | linear model | 900T0C | |
| LPV | LPV model | linear models:  SS0T100A, SS0T100B, SS0T100C, SS100T0A, SS100T0B, SS100T0C, SS0T450A, SS0T450B, SS0T450C, SS450T0A, SS450T0B, SS450T0C, SS0T900A, SS0T900B, SS0T900C, SS900T0A, SS900T0B, SS900T0C | curve fitting:  $A$: second-order polynomial  $B$: second-order polynomial  $C$: second-order polynomial  $D$: linear function  $BC$: two-term exponential |
| nonlinear A | nonlinear HW model | MULTI | input nonlinearity: third-order polynomial  output nonlinearity: unit gain (absent)  OE model order: $n_b = 2$, $n_f = 1$, $n_k = 0$ |
| nonlinear B | nonlinear HW model | VARIOUS | input nonlinearity: third-order polynomial  output nonlinearity: unit gain (absent)  OE model order: $n_b = 2$, $n_f = 1$, $n_k = 0$ |

[a]The data location is tabulated in electronic supplementary material, table S1.

In contrast with the linear model, the nonlinear models were identified from the data with multiple steps under varying conditions. In table 3, model identification data for all developed models are tabulated.

# 4. Results and discussion

In electronic supplementary material, table S2, the LTI model parameters are shown. From these LTI models, the LPV model was developed. The correlations of the model parameters were constructed via

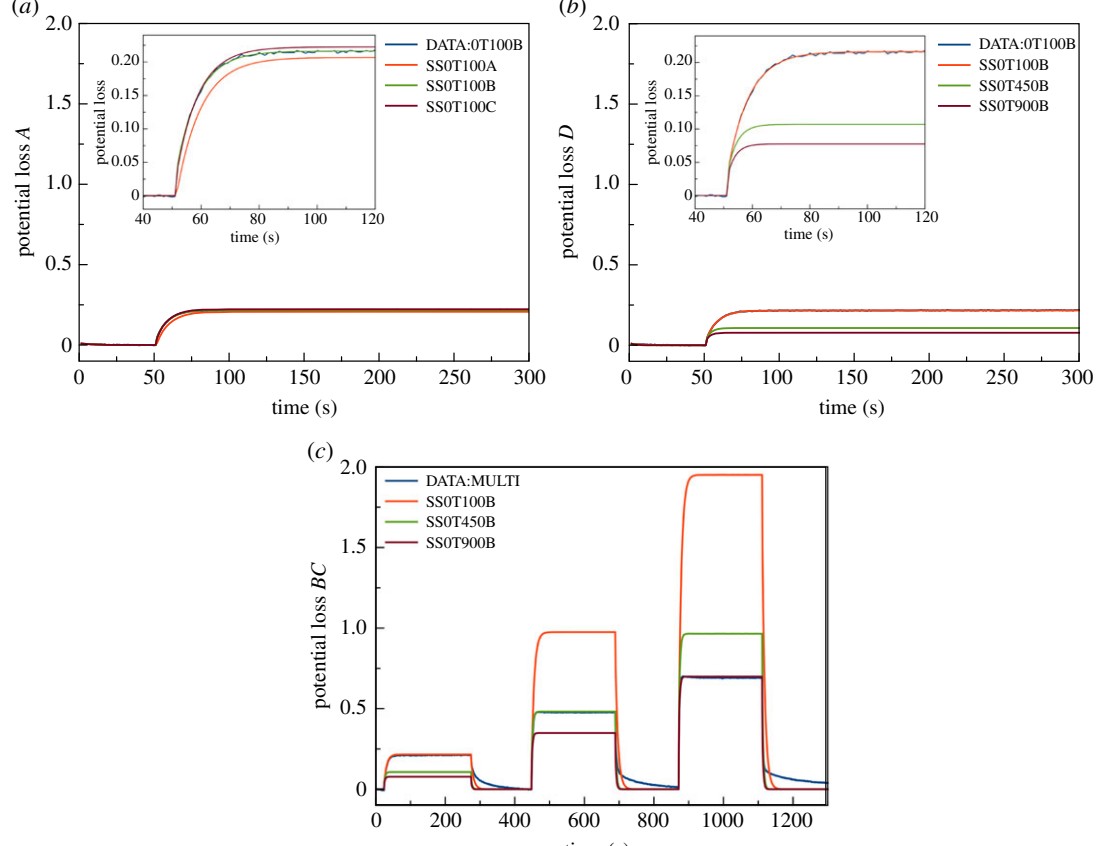

**Figure 4.** Comparison of response between measured data (blue lines) and predicted data from linear models: (*a*) matching conditions with current step from 0 to 100 mA, (*b*) different conditions with current step from 0 to 100 mA and (*c*) multiple current steps from 0 to 100, 450 and 900 mA.

curve fitting. The nonlinear models were identified from the multiple step and various step data [40]. A third-order polynomial function was selected for the Hammerstein block. The model order of the OE model was $n_b = 2$, $n_f = 1$ and $n_k = 0$. In table 3, the identifying conditions are summarized. In addition, parameter values estimated in this work are provided in electronic supplementary material, table S3.

In the following sections, the graphical highlights of validation and comparison results are displayed. Full graphical results of the linear model and the LPV model are shown in electronic supplementary material, figures S11 and S12, respectively. In electronic supplementary material, table S5, the fit percentage values of the prediction results are tabulated.

## 4.1. Linear state-space model

The linear models were identified as first-order state-space models. The number of states ($n$) was one. Model parameters were estimated using one set of experimental data. To validate the models, different sets of experimental data were applied. In figure 4, validation results for the linear models are shown. Figure 4*a* shows the validation results with the same conditions (current steps) as used in the estimation (0–100 mA). Results demonstrate that the models were able to accurately predict individual response data.

Figure 4*b* highlights the results when the models were validated at different conditions (different current steps). It was found that the models could predict accurately only the data used to identify the models' parameters. The models poorly estimated other data. The gain of the models significantly deviated. Results suggested that the linear model was only accurate locally.

Figure 4*c* provides an example by displaying a comparison between model predictions and measured data in the context of multiple step current discharges. Results clarified the dependency between gain and current level. Thus, from the results shown, the linear models were able to accurately predict the responses if the current level corresponded with the models. Nevertheless, most of the battery data contain more than

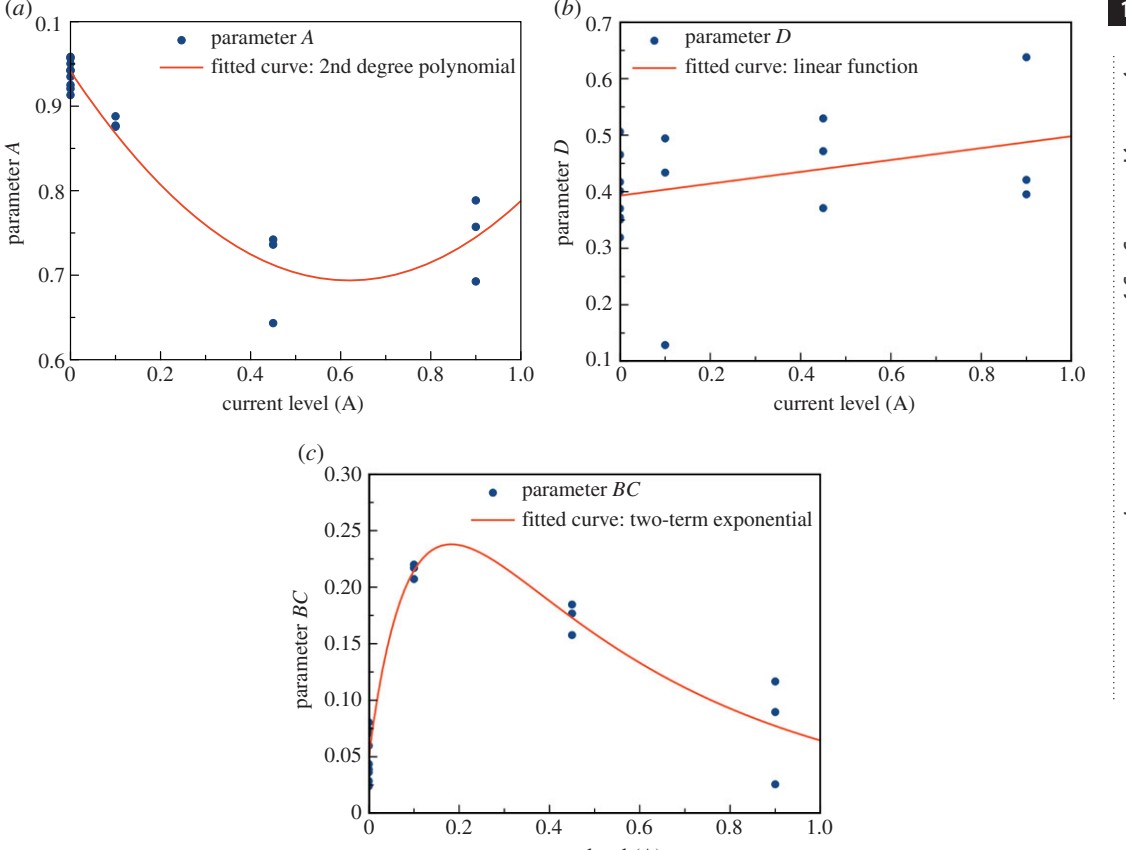

**Figure 5.** Correlations of state space model parameters as functions of current levels: (*a*) parameter *A*, (*b*) parameter *D* and (*c*) parameter *BC*.

one current level. Consequently, the linear model cannot be applied in most cases. In this situation, the LPV model proved to offer the level of flexibility necessary for adapting to the LTI responses.

## 4.2. LPV model

As previously stated, the LPV model was developed from linear state-space models. Consequently, the models with different current level conditions were combined into one model. For this model, the conditions of current level included: 0, 100, 450 and 900 mA. Each condition, with respect to the final current level, provided a different set of model parameters. For instance, the state-space model estimated from the current step of 0 to 100 mA provided the values of model parameters at the 100 mA current level. At each current level, three datasets of the same condition were used. For validation, the LPV model was then used to predict the various response data.

As shown in figure 5, correlations between model parameters and current levels were fitted in accordance with equations (3.4) and (3.5). In figure 5*a,b*, the correlation of model parameters *A* and *D* was able to be fitted using a second-order polynomial function as well as a linear function, respectively. Parameters *A* and *D* showed consistent trends with respect to current levels. However, parameters *B* and *C* were found to be inconsistent in their trends. Moreover, the values of *B* and *C* contained both positive and negative values which can cause discrepancy in prediction. To address this issue, *C* was fixed at *C* = 1, while *B* and *C* were multiplied together, resulting in the parameter *BC* which proved to be more consistent, as described in equations (3.6) and (3.7). Accordingly, the LPV model becomes

$$X(k+1) = A(p)X(k) + BC(p)u(k) \qquad (4.1)$$

and

$$Y(k) = X(k) + D(p)u(k). \qquad (4.2)$$

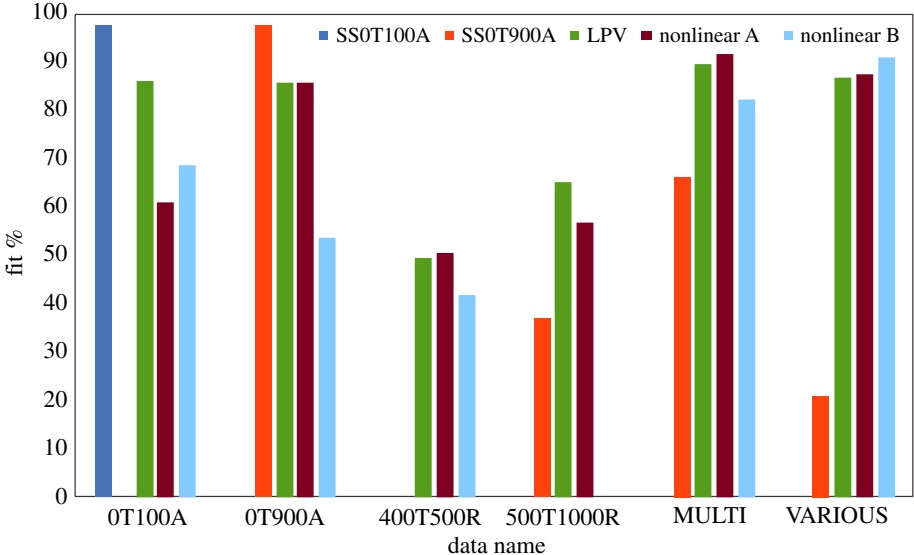

**Figure 6.** Comparison of fit percentage of model prediction between various models and data. Fit % indicates how well the model prediction fits the estimation data and expressed as: $100(1 - ((y - \hat{y})/(y - \text{mean}(y))))$.

The trend of parameter $BC$ exhibited good consistency and was able to be fitted via a two-term exponential function, as shown in figure 5$c$. For physical interpretation, equations (4.1) and (4.2) are equivalent to the ECM as expressed in equations (3.8) and (3.9). Parameter $D$ was expressed as a linear function having a small slope. This indicated that the discharge current level had little effect on $R_0$. Parameters $A$ and $BC$ were fitted with a polynomial and exponential function, respectively. As the RC loop represented the activation loss, these correlations agreed with the nonlinear trend of the activation loss.

As regards validation, the developed LPV model was used to predict the same response data as used previously in §4.1. A comparison of the fit percentage between various model predictions is shown in figure 6. As for the single step responses (0T100A and 0T900A), results demonstrated good agreement between measured data and predicted data. Compared with the linear model, however, the LPV model proved to be slightly less accurate due to the error in correlation fitting. Yet, the LPV model performed much better globally because the models used for constructing this LPV model were estimated from data measured directly. In addition to the LPV model, two nonlinear models, nonlinear A and nonlinear B (table 3), were identified and compared for response prediction. Results showed that the LPV model performed better than the nonlinear models in this case. The nonlinear models, identified from the data, were seen to have high complexity. Thereby, the models were found to be less robust (especially nonlinear model B).

As regards multiple step responses (MULTI), prediction results of the multiple step responses are displayed in figure 7$a$. Results highlighted the benefit of the LPV model revealing that the LPV model was able to predict multiple step responses with acceptable agreement. In comparison with the linear model, the LPV model confirmed improvement in prediction. In addition, when the current level changed, the LPV model was able to predict cell voltage more accurately than the linear model. The gain of the LPV model prediction was able to adapt to current level change. The LPV model proved to be comparable with that of the nonlinear model identified from the matching data (nonlinear A). However, the nonlinear model identified from the other condition (nonlinear B) indicated less accurate prediction.

For validation purposes, the LPV model was tested further, using the different sets of data that had not been used for estimation of the coefficients in the underlying LTI models. The measured data with the repeating step currents: 400–500 mA (400T500R) and 500–1000 mA (500T1000R) were used for validation. A comparison of the fit percentages found that all the proposed models including the LPV model and nonlinear models were less accurate than the other datasets in predicting the responses. As shown in electronic supplementary material, figure S12, the response comparison revealed two limitations of the LPV model: the effect of SOC and the input range of the underlying LTI models.

Regarding the effect of SOC, the error of prediction increased as time passed because cell voltage is also a function of SOC [14]. As the battery discharged over time, cell voltage dropped because of the

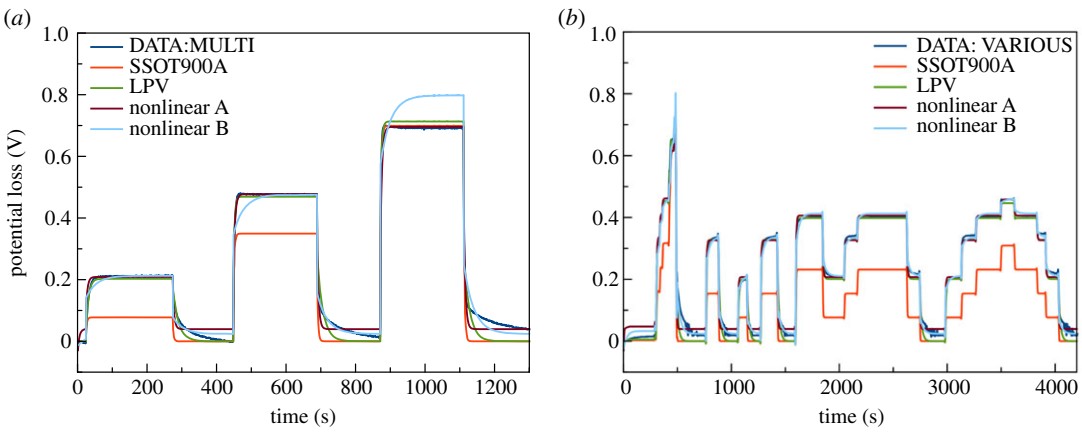

**Figure 7.** Comparison of response between measured data (dark blue lines), predicted data from linear model (red line), LPV model (green line) and nonlinear models: (*a*) multiple current steps from 0 to 100, 450 and 900 mA and (*b*) various current steps.

decrease in SOC. However, in this model, the effect of SOC on cell voltage was not considered. Another limitation shown is regarding the input range of the underlying LTI model. For instance, the upper bound of the current level of the underlying LTI models was 900 mA. For the current level higher than 900 mA, the correlation of the model parameters was found to be incorrect; less accurate values were obtained from extrapolation. Thus, this indicated that the LPV model was not precise in predicting conditions which are out of the input range of the underlying LTI models.

In figure 7*b*, prediction results of the various step responses (VARIOUS) are displayed, verifying the models against more complicated data. Limitation of the model appeared the same as in a previous test where cell voltage is dependent on SOC. Nonetheless, the LPV model exhibited superior performance when compared with the linear model and its performance was comparable to that of the nonlinear model. This result revealed the feasibility of using the LPV model. The LPV approach sets out to prove its significance as a modelling tool for the nonlinear behaviour of a ZAB. Herein, the discharge current level is demonstrated as the effective scheduling parameter for predicting the nonlinear behaviour of a ZAB. For some large-scale refuellable ZABs, the influence of SOC is less concerned. Hence, the management system having only discharge current scheduling might be viable over a wide range of operations. To improve the LPV model, the model may have to be developed further by including other scheduling parameters such as SOC or temperature. Moreover, it might be feasible to study the LPV model in a rechargeable ZAB, as the charging process of this battery also adopts the nonlinear characteristic.

## 5. Conclusion

In this work, an LPV model was developed to predict the nonlinear dynamic behaviour of a ZAB. LTI models were used as the basis to construct the LPV model. The experimental data acquired from an in-house designed tubular refuellable ZAB were used for identification purposes and validation. By comparing model accuracy based on normalized root mean square error, results showed that the linear model, identified at each local point, was able to predict the behaviour of a ZAB but only at the local vicinity of that point. However, it was unable to capture the nonlinear behaviour of the ZAB where the gain intensely varied with the discharge current levels. In contrast, the LPV model could well predict battery response. Further, the LPV model was found to be more robust than two other nonlinear models. The LPV model sets out to prove its worth as a dynamic modelling approach for a ZAB.

Data accessibility. Electronic supplementary material is provided. The datasets and supplementary file have been uploaded to The Open Science Framework at https://osf.io/mnbpg/, and the DOI is 10.17605/OSF.IO/MNBPG.
Authors' contributions. Conceptualization: S.K.; methodology: W.L. and S.K.; investigation: W.L.; formal analysis: W.L., S.O. and S.K.; writing—original draft preparation: W.L. and S.K.; writing—review and editing: W.L., S.O., S.D., S.S., A.A., R.C. and S.K.; funding acquisition: S.O. and S.K. All authors have read and agreed to the published version of the manuscript.
Competing interests. The authors declare no potential conflict of interest.
Funding. W.L. thanks the Dusadeepipat scholarship. This research was supported by the Program Unit for Human Resources & Institutional Development, Research and Innovation – CU (grant no. B16F630071) with C2F Global Partnership Program, the Energy Storage Cluster, Chulalongkorn University and the iCODE Institute, research

project of the IDEX Paris-Saclay, and by the Hadamard Mathematics LabEx (LMH) through grant no. ANR-11-LABX-0056-LMH in the 'Programme des Investissements d'Avenir'. S.S. thanks Ratchadaphiseksomphot Endowment Fund, Chulalongkorn University, for financial support for his stay in Bangkok.

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
