## [Reviewer comments · Royal Society Open Science]

Review History

RSOS-201107.R0 (Original submission)

Review form: Reviewer 1

Is the manuscript scientifically sound in its present form?

Yes

Are the interpretations and conclusions justified by the results?

No

Is the language acceptable?

Yes

Do you have any ethical concerns with this paper?

No

Have you any concerns about statistical analyses in this paper?

No

Recommendation?

Major revision is needed (please make suggestions in comments)

Comments to the Author(s)

In this manuscript, the authors proposed an LPV model to forecast the nonlinear dynamic behavior of a ZAB. They also used experimental designed tubular refuellable ZAB to verify their modeling. They found that their LPV model is more robust than other models. It is an interesting manuscript. However, I have some concerns.

1. If possible, please tight the manuscript
2. Are the experimental data from Lao-atiman's paper? Or only some of them from Lao-atiman's paper?
3. Did the author get the general parameters for the LPV model for ZAB?
4. With the fitted parameters and the proposed LPV model, is it possible to predict dynamic behavior of a ZAB?
5. Introduction "The overall reactions that occur in the battery are described as follows:....." Can you provide some references to support this statement (i.e. the overall reactions)?
6. Introduction "The theoretical open circuit voltage is 1.65 V. However, the practical OCV obtained from laboratory prototypes is about 1.4 V." Please provide the equation to calculate the theoretical OCV, and provide the references for the OCV from laboratory.
7. Introduction "Alternatively, nonlinear behavior can be captured via a linear parameter varying (LPV) model, which approximates a nonlinear system with high accuracy." Please provide references to support this statement.

Review form: Reviewer 2

Is the manuscript scientifically sound in its present form?

Yes

Are the interpretations and conclusions justified by the results?

Yes

Is the language acceptable?

Yes

Do you have any ethical concerns with this paper?

No

Have you any concerns about statistical analyses in this paper?

No

Recommendation?

Major revision is needed (please make suggestions in comments)

Comments to the Author(s)

The authors present a well conducted and highly readable study on the modeling of zinc-air batteries. They introduce nicely into the topic and underline the necessity for a dynamic model. The mathematical derivations are sound.

There is one big shortcoming so that a major revision is needed: I am not sure whether the authors made the right choice of simulations to show the benefit of the nonlinear model. All simulations show input / response times in the range of several hundreds or tens of seconds. However, if the dynamic behavior can be elucidated as claimed, then also pulse current operation (resting time and pulse time in the range of 0.5 s) must be included in the simulations.

The pulse-current operation of zinc air batteries is a highly topic operation mode - e.g. to prevent dendrite formation at the anode or to ensure oxygen supply at the gas-diffusion-electrode - which should also be mentioned by appropriate literature in the introduction.

Moreover, the figures are not very appealing for the reader (too many plots chosen; many of the figures look the same so that it is difficult to get a message). Please select fewer but more meaningful plots and improve the visual appeal of them.

Decision letter (RSOS-201107.R0)

Dear Professor Kheawhom:

Title: Linear Parameter Varying Model for a Refuellable Zinc-air Battery
Manuscript ID: RSOS-201107

The editor assigned to your manuscript has now received comments from reviewers. I apologise that this has taken longer than usual. We would like you to revise your paper in accordance with the referee and Subject Editor suggestions which can be found below (not including confidential reports to the Editor). Please note this decision does not guarantee eventual acceptance.

Please submit your revised paper before 15-Oct-2020. Please note that the revision deadline will expire at 00.00am on this date. If we do not hear from you within this time then it will be assumed that the paper has been withdrawn. In exceptional circumstances, extensions may be possible if agreed with the Editorial Office in advance. We do not allow multiple rounds of revision so we urge you to make every effort to fully address all of the comments at this stage. If deemed necessary by the Editors, your manuscript will be sent back to one or more of the original reviewers for assessment. If the original reviewers are not available we may invite new reviewers.

Yours sincerely,
Dr Laura Smith

Publishing Editor, Journals

On behalf of the Subject Editor Professor Anthony Stace and the Associate Editor Professor Kim Jelfs.

RSC Associate Editor:
 Comments to the Author:
 Please address carefully the reviewer's comments.

RSC Subject Editor:
 Comments to the Author:
 (There are no comments.)

Reviewers' Comments to Author:
 Reviewer: 1

Comments to the Author(s)

In this manuscript, the authors proposed an LPV model to forecast the nonlinear dynamic behavior of a ZAB. They also used experimental designed tubular refuellable ZAB to verify their modeling. They found that their LPV model is more robust than other models. It is an interesting manuscript. However, I have some concerns.

1. If possible, please tight the manuscript
2. Are the experimental data from Lao-atiman's paper? Or only some of them from Lao-atiman's paper?
3. Did the author get the general parameters for the LPV model for ZAB?
4. With the fitted parameters and the proposed LPV model, is it possible to predict dynamic behavior of a ZAB?
5. Introduction "The overall reactions that occur in the battery are described as follows:....." Can you provide some references to support this statement (i.e. the overall reactions)?
6. Introduction "The theoretical open circuit voltage is 1.65 V. However, the practical OCV obtained from laboratory prototypes is about 1.4 V." Please provide the equation to calculate the theoretical OCV, and provide the references for the OCV from laboratory.
7. Introduction "Alternatively, nonlinear behavior can be captured via a linear parameter varying (LPV) model, which approximates a nonlinear system with high accuracy." Please provide references to support this statement.

Reviewer: 2

Comments to the Author(s)

The authors present a well conducted and highly readable study on the modeling of zinc-air batteries. They introduce nicely into the topic and underline the necessity for a dynamic model. The mathematical derivations are sound.

There is one big shortcoming so that a major revision is needed: I am not sure whether the authors made the right choice of simulations to show the benefit of the nonlinear model. All

simulations show input / response times in the range of several hundreds or tens of seconds. However, if the dynamic behavior can be elucidated as claimed, then also pulse current operation (resting time and pulse time in the range of 0.5 s) must be included in the simulations. The pulse-current operation of zinc air batteries is a highly topic operation mode - e.g. to prevent dendrite formation at the anode or to ensure oxygen supply at the gas-diffusion-electrode - which should also be mentioned by appropriate literature in the introduction.

Moreover, the figures are not very appealing for the reader (too many plots chosen; many of the figures look the same so that it is difficult to get a message). Please select fewer but more meaningful plots and improve the visual appeal of them.

Author's Response to Decision Letter for (RSOS-201107.R0)

See Appendix A.

RSOS-201107.R1 (Revision)

Review form: Reviewer 1

Is the manuscript scientifically sound in its present form?

Yes

Are the interpretations and conclusions justified by the results?

Yes

Is the language acceptable?

Yes

Do you have any ethical concerns with this paper?

No

Have you any concerns about statistical analyses in this paper?

No

Recommendation?

Accept as is

Comments to the Author(s)

The authors have considered all points I raised in my initial report. The manuscript is improved. I recommend the publication of this manuscript

Decision letter (RSOS-201107.R1)

Dear Professor Kheawhom:

Title: Linear Parameter Varying Model for a Refuellable Zinc-air Battery
Manuscript ID: RSOS-201107.R1

It is a pleasure to accept your manuscript in its current form for publication in Royal Society Open Science. The chemistry content of Royal Society Open Science is published in collaboration with the Royal Society of Chemistry.

On behalf of the Subject Editor Professor Anthony Stace and the Associate Editor Professor Kim Jelfs.

RSC Associate Editor:
Comments to the Author:
(There are no comments.)

RSC Subject Editor:
Comments to the Author:
(There are no comments.)

Reviewer(s)' Comments to Author:
Reviewer: 1

Comments to the Author(s)
The authors have considered all points I raised in my initial report. The manuscript is improved. I recommend the publication of this manuscript

Appendix A

Reviewer: 1

Comments to the Author(s)

In this manuscript, the authors proposed an LPV model to forecast the nonlinear dynamic behavior of a ZAB. They also used experimentally designed tubular refuellable ZAB to verify their modeling. They found that their LPV model is more robust than other models. It is an interesting manuscript. However, I have some concerns.

1. If possible, please tighten the manuscript

The manuscript has been revised. The number of plots has been reduced for tightening the manuscript. Some plots were cut and moved to the supplementary file instead. The remaining Figs. were meaningful and necessary for discussion.

2. Are the experimental data from Lao-atiman's paper? Or only some of them from Lao-atiman's paper?

Some of the data are from Lao-atiman's paper. Only data named "MULTI" and "VARIOUS" are not from Lao-atiman's paper. Details of data location are indicated in Table S1 in the supplementary file.

3. Did the author get the general parameters for the LPV model for ZAB?

No, the parameters are not general. The LPV parameters obtained can only be used for the specifically fabricated ZAB similar to this ZAB.

4. With the fitted parameters and the proposed LPV model, is it possible to predict dynamic behavior of a ZAB?

With the LPV parameters fitted in this work, it can only be used to predict the behavior of the ZAB in this work or a ZAB fabricated the same as in this work. For other batteries, the parameters have to be fitted for each individual battery.

5. Introduction "The overall reactions that occur in the battery are described as follows:....." Can you provide some references to support this statement (i.e. the overall reactions)?

The references to support this statement have been added to the manuscript.

10 Lee, J.-S., Tai Kim, S., Cao, R., Choi, N.-S., Liu, M., Lee, K. T., Cho, J. 2011 Metal–Air Batteries with High Energy Density: Li–Air versus Zn–Air. *Advanced Energy Materials*. 1, 34-50. (<https://doi.org/10.1002/aenm.201000010>)

11 Li, Y., Dai, H. 2014 Recent advances in zinc-air batteries. *Chemical Society Reviews*. 43, 5257-5275. (<https://doi.org/10.1039/C4CS00015C>)

6. Introduction "The theoretical open circuit voltage is 1.65 V. However, the practical OCV obtained from laboratory prototypes is about 1.4 V." Please provide the equation to calculate the theoretical OCV, and provide the references for the OCV from laboratory.

This part of the introduction has been revised. The OCV calculation equation along with its description has been added. References for the practical OCV have also been provided.

- 12 Wang, X., Sebastian, P. J., Smit, M. A., Yang, H., Gamboa, S. A. 2003 *Studies on the oxygen reduction catalyst for zinc-air battery electrode*. *Journal of Power Sources*. 124, 278-284. ([https://doi.org/10.1016/S0378-7753\(03\)00737-7](https://doi.org/10.1016/S0378-7753(03)00737-7))
- 13 Zhu, S., Chen, Z., Li, B., Higgins, D., Wang, H., Li, H., Chen, Z. 2011 *Nitrogen-doped carbon nanotubes as air cathode catalysts in zinc-air battery*. *Electrochimica Acta*. 56, 5080-5084. (<https://doi.org/10.1016/j.electacta.2011.03.082>)
- 14 Larsson, F., Rytinki, A., Ahmed, I., Albinsson, I., Mellander, B.-E. 2017 *Overcurrent Abuse of Primary Prismatic Zinc-Air Battery Cells Studying Air Supply Effects on Performance and Safety Shut-Down*. *Batteries*. 3, 1. (<https://doi.org/10.3390/batteries3010001>)

7. Introduction “Alternatively, nonlinear behavior can be captured via a linear parameter varying (LPV) model, which approximates a nonlinear system with high accuracy.” Please provide references to support this statement.

The references to support this statement have been added to the manuscript.

- 35 Tóth, R., Heuberger P.S.C., Van den Hof, P.M.J. 2012 *Prediction-Error Identification of LPV Systems: Present and Beyond*. In *Control of Linear Parameter Varying Systems with Applications* (eds J. Mohammadpour, C. Scherer), pp. 27–58. New York, USA: Springer Science+Business Media.
- 36 Schoukens, M., Tóth, R. 2018 *From Nonlinear Identification to Linear Parameter Varying Models: Benchmark Examples*. *IFAC-PapersOnLine*. 51, 419-424. (<https://doi.org/10.1016/j.ifacol.2018.09.181>)

Reviewer: 2

Comments to the Author(s)

The authors present a well conducted and highly readable study on the modeling of zinc-air batteries. They introduce nicely into the topic and underline the necessity for a dynamic model. The mathematical derivations are sound.

There is one big shortcoming so that a major revision is needed: I am not sure whether the authors made the right choice of simulations to show the benefit of the nonlinear model. All simulations show input / response times in the range of several hundreds or tens of seconds. However, if the dynamic behavior can be elucidated as claimed, then also pulse current operation (resting time and pulse time in the range of 0.5 s) must be included in the simulations. The pulse-current operation of zinc air batteries is a highly topic operation mode - e.g. to prevent dendrite formation at the anode or to ensure oxygen supply at the gas-diffusion-electrode - which should also be mentioned by appropriate literature in the introduction.

Moreover, the figures are not very appealing for the reader (too many plots chosen; many of the figures look the same so that it is difficult to get a message). Please select fewer but more meaningful plots and improve the visual appeal of them.

The manuscript has been revised. Three Figs. have been revised to make the paper more concise.

Fig.4 has been revised. The number of plots within the Fig. has been reduced from 6 to 3. Now, the Fig. only shows the highlight comparison results with meaningful discussion.

Fig.5 has been revised. The number of plots within the Fig. has been reduced from 5 to 3.

Fig.6 has been changed. New Fig. 6 is a bar graph comparing the fit percentage. The old Fig. 6 has been changed to the new Fig. 7 with only 2 highlight plots left.

All cut Figs. have been moved to the supplementary file instead.

As for the pulse-current operation mentioned by the reviewer, we might try the pulse current in the ZAB with a different configuration in the future. The pulse-current operation is not included in this work because this configuration of ZAB is a refuellable battery which is only capable of discharge and cannot be charged. For a ZAB, charging with pulse-current is a technique used to prevent the growth of zinc dendritic morphology in the charging process. However, there is no research in the literature concerning the pulse current for discharge of a ZAB. Moreover, there is no actual application for pulse current discharge of a ZAB. Thus, the pulse-current discharge has not been included in this work.

The manuscript has been revised according to this comment. The information of pulsating current charging has been added to the introduction with appropriate references.